# Accurate detection of m$^6$A RNA modifications in native RNA sequences

Huanle Liu[1,2,11], Oguzhan Begik [1,2,3,11], Morghan C. Lucas [1,4], Jose Miguel Ramirez[1], Christopher E. Mason [5,6,7], David Wiener[8], Schraga Schwartz[8], John S. Mattick[2,3,10], Martin A. Smith [3,9] & Eva Maria Novoa [1,2,3,4]

The epitranscriptomics field has undergone an enormous expansion in the last few years; however, a major limitation is the lack of generic methods to map RNA modifications transcriptome-wide. Here, we show that using direct RNA sequencing, $N^6$-methyladenosine (m$^6$A) RNA modifications can be detected with high accuracy, in the form of systematic errors and decreased base-calling qualities. Specifically, we find that our algorithm, trained with m$^6$A-modified and unmodified synthetic sequences, can predict m$^6$A RNA modifications with ~90% accuracy. We then extend our findings to yeast data sets, finding that our method can identify m$^6$A RNA modifications in vivo with an accuracy of 87%. Moreover, we further validate our method by showing that these 'errors' are typically not observed in yeast *ime4*-knockout strains, which lack m$^6$A modifications. Our results open avenues to investigate the biological roles of RNA modifications in their native RNA context.

[1] Centre for Genomic Regulation (CRG), The Barcelona Institute of Science and Technology, 08003 Barcelona, Spain. [2] Department of Neuroscience, Garvan Institute of Medical Research, Darlinghurst, New South Wales 2010, Australia. [3] St-Vincent's Clinical School, UNSW Sydney, Darlinghurst, New South Wales 2010, Australia. [4] Universitat Pompeu Fabra (UPF), Barcelona, Spain. [5] Department of Physiology and Biophysics, Weill Cornell Medicine, New York, NY 10021, USA. [6] The Feil Family Brain and Mind Institute, Weill Cornell Medicine, New York, NY 10021, USA. [7] The WorldQuant Initiative for Quantitative Prediction, Weill Cornell Medicine, New York, NY 10021, USA. [8] Department of Molecular Genetics, Weizmann Institute of Science, Rehovot, Israel. [9] Kinghorn Centre for Clinical Genomics, Garvan Institute of Medical Research, Darlinghurst, New South Wales 2010, Australia. [10] Present address: Green templeton College, Oxford OX2 6HG, UK. [11] These authors contributed equally: Huanle Liu, Oguzhan Begik. Correspondence and requests for materials should be addressed to E.M.N. (email: eva.novoa@crg.eu)

In the last few years, our ability to map RNA modifications transcriptome-wide has revolutionized our understanding of how these chemical entities shape cellular processes, modulate cancer risk, and govern cellular fate[1–4]. Systematic efforts to characterize this regulatory layer have revealed that RNA modifications are far more widespread than previously thought, can be subjected to dynamic regulation, and can profoundly impact RNA processing stability and translation[5–10]. A fundamental challenge in the field, however, is the lack of a generic approach for mapping and quantifying RNA modifications, as well as, the lack of single molecule resolution[11].

Current technologies to map the epitranscriptome rely on next-generation sequencing and, as such, they are typically blind to nucleotide modifications. Consequently, indirect methods are required to identify RNA modifications transcriptome-wide, which has been mainly approached using two different strategies: (i) antibody immunoprecipitation, which specifically recognizes the modified ribonucleotide[5,6,12–14]; and (ii) chemical-based detection, using chemical compounds that selectively react with the modified ribonucleotide of interest, followed by reverse-transcription of the RNA fragment, which leads to accumulation of reads that have the same identical ends[8,9,15]. Although these methods have provided highly valuable information, they are limited by the available repertoire of commercial antibodies and the lack of selective chemical reactivities towards a particular RNA modification[16], often lack single nucleotide resolution[5–7] or require complex protocols to achieve it[17], cannot provide quantitative estimates of the stoichiometry of the modification at a given site, and are often unable to identify the underlying RNA molecule that is modified.

To overcome these limitations, third-generation sequencing technologies, such as the platforms provided by Oxford Nanopore Technologies (ONT)[18] and Pacific Biosciences (PacBio)[3], have been proposed as a new means to detect RNA modifications in native RNA sequences. RNA modifications can be detected using PacBio in the form of kinetic changes of reverse transcriptases, which occur when the enzyme encounters a modified RNA nucleotide[19]. On the other hand, RNA modifications can be identified in its native RNA context using ONT, by pulling the RNA molecules through the nanopores that are embedded in a membrane. The ONT platform relies on the measurement of disruptions in the current intensity as the RNA or DNA molecule passes through the pore, which can be used to identify the transiting nucleotides. Although ONT direct RNA sequencing is already a reality[20,21], extracting RNA modification information from ONT reads is still an unsolved challenge. RNA modifications are known to cause disruptions in the pore current that can be detected upon comparison of raw current intensities — also known as "squiggles"[18,20]. However, current efforts have not yet yielded an efficient and accurate RNA modification detection algorithm, largely due to the challenges in the alignment and re-squiggling of RNA current intensities[22,23].

As an alternative strategy, we hypothesized that the current intensity changes caused by the presence of RNA modifications may lead to increased "errors" and decreased qualities from the output of base-calling algorithms that do not model base modifications (Fig. 1a). Indeed, here we find that base-calling "errors" can accurately identify $N^6$-methyladenosine (m6A) RNA modifications in native RNA sequences, and propose a novel algorithm, EpiNano (github.com/enovoa/EpiNano), which can be used to identify m6A RNA modifications from RNA reads with an overall accuracy of ~90%. Our results provide a proof of concept for the use of base-called features to identify RNA modifications using direct RNA sequencing, and open avenues to explore additional RNA modifications in the future.

## Results

### RNA modifications alter base-called features in ONT reads.
Previous work has shown that ONT raw current intensity signals, known as "squiggles", can be subdivided into "events", which correspond to consecutive 5-mer sequences shifted one nucleotide at a time (e.g., in the sequence AGACAAU, the corresponding 5-mer "events" would be AGACA, GACAA, and ACAAU)[24–27]. Therefore, to systematically identify the current intensity changes caused by the presence of a given RNA modification, perturbations of the current intensity signals should be measured and analyzed for each possible 5-mer ($n = 1024$).

To this end, we designed a set of synthetic sequences that comprised all possible 5-mers (median occurrence of each 5-mer = 10), while minimizing the RNA secondary structure (see the Methods section and Supplementary Note 1). We then employed direct RNA sequencing to characterize the differences of in vitro transcribed (IVT) constructs that incorporated either "m6A" instead of adenine, or unmodified ribonucleotides (Fig. 1a, Supplementary Table 1 for sequencing metrics). Comparison of the two data sets revealed that base-called m6A-modified reads were significantly enriched in mismatches compared with their unmodified counterparts (Fig. 1b, c), and that these "errors" were mainly, but not exclusively, located in adenine positions. We observed that, in addition to mismatch frequency, other metrics including per-base quality, insertion frequency, deletion frequency, and current intensity were significantly altered (Fig. 1c and Supplementary Fig. 1). Moreover, these "errors" were highly reproducible in independent biological replicates with respect to mismatch frequency, deletion frequency, per-base quality, and current intensity (Fig. 1d–g). By contrast, insertion frequencies were not reproducible across biological replicates (Supplementary Fig. 1), suggesting that this feature is likely unrelated to the presence of RNA modifications, and thus was not further considered in downstream analyses.

### Base-calling "errors" can accurately predict m6A modifications.
We then examined whether these observed differences would be sufficient to accurately classify a given site into "modified" or "unmodified". For this aim, we first focused our analysis on 5-mers that matched the known m6A motif RRACH, as these would be the most relevant in which to detect m6A modifications. To reveal whether the features from m6A-modified RRACH k-mers were distinct from unmodified RRACH k-mers, we compiled the base-called features (base quality, mismatch frequency, and deletion frequency) for each position of the k-mer (−2, −1, 0, +1, and +2) (Fig. 2a, Supplementary Fig. 2), and performed principal component analysis of the features, finding that the two populations (m6A-modified and unmodified RRACH k-mers) were largely nonoverlapping (Fig. 2b). As a control, we performed the same analysis in k-mers with identical sequence context, but centered in C, G, or U (instead of A), finding that no differences could be observed between these populations (Fig. 2c), suggesting that the observed differences are m6A-specific, and not data set-specific.

To statistically determine whether these features could be used to accurately classify a given site into "m6A-modified" or "unmodified", we trained multiple support vector machines (SVM) using as input the base-called features from m6A-containing RRACH k-mers and unmodified RRACH k-mers (see the Methods section). We first tested whether each individual feature at position 0 (the modified site) was able to classify a given RRACH k-mer into m6A-modified or unmodified. Our results show that base quality, deletion frequency and mismatch frequency alone were able to accurately predict the modification status with reasonable accuracy (70–86% accuracy, depending on

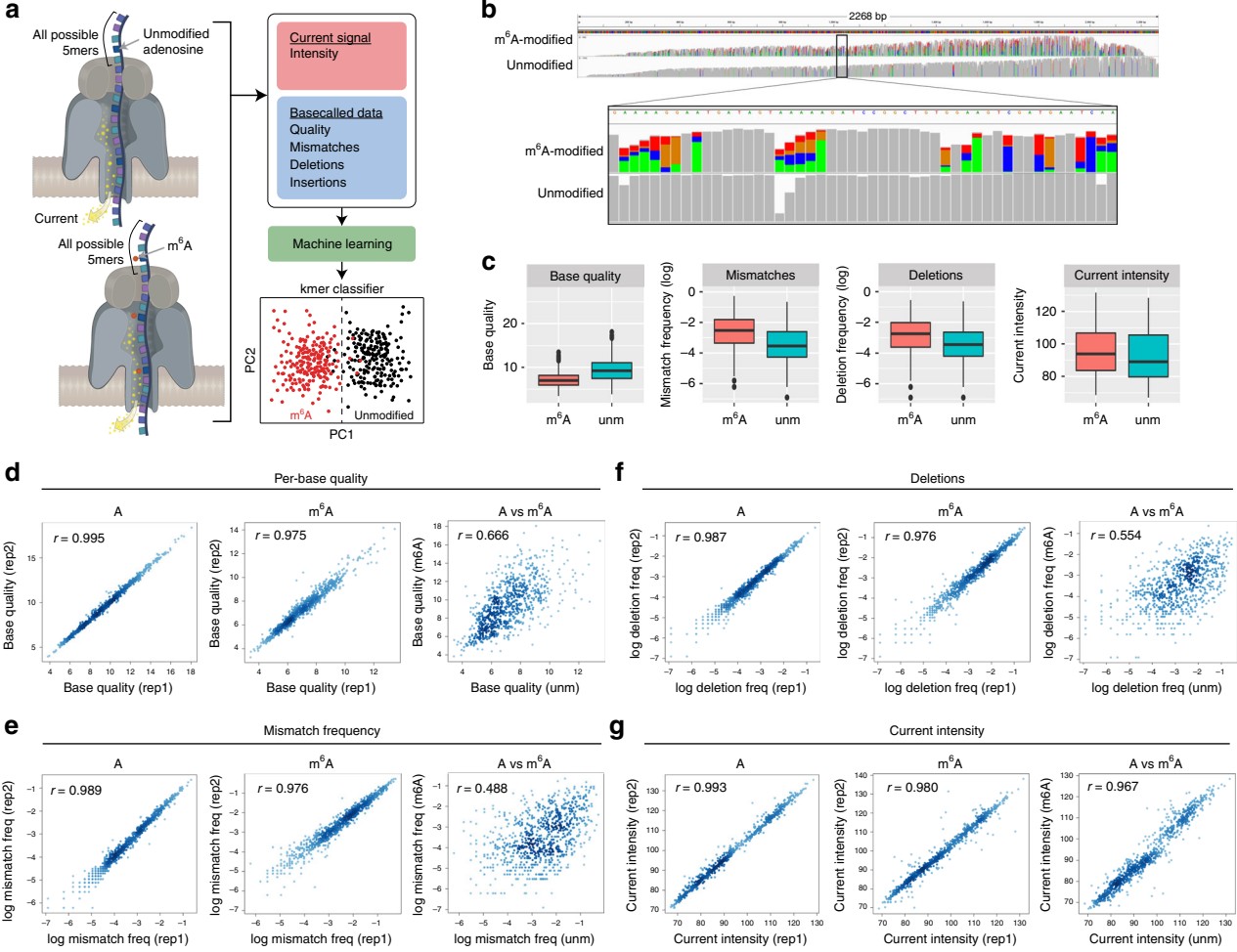

**Fig. 1** Base-calling "errors" can be used as a proxy to identify RNA modifications in direct RNA sequencing reads. **a** Schematic overview of the strategy used in this work to train and test an m6A RNA base-calling algorithm. **b** IGV snapshot of one of the four transcripts used in this work. In the upper panel, in vitro transcribed reads containing m6A have been mapped, whereas in the lower panel the unmodified counterpart is shown. Nucleotides with mismatch frequencies >0.05 have been colored. **c** Comparison of m6A and A positions, at the level of per-base quality scores (left panel), mismatch frequencies (middle left panel), deletion frequency (middle right panel), and mean current intensity (right panel). All possible k-mers (computed as a sliding window along the transcripts) have been included for these comparisons (n = 9974). **d**–**g** Replicability of each individual feature — base quality (**d**), deletion frequency (**f**), mismatch frequency (**e**), and current intensity (**g**) — across biological replicates, for both unmodified ("A") and m6A-modified ("m6A") data sets. Comparison of unmodified and m6A-modified ("A vs m6A") is also shown for each feature. Correlation values shown correspond to Spearman's rho. Error bars indicate s.d. Source data are provided in the Source Data file

the feature used) (Fig. 2d, see also Supplementary Table 2 and the Methods section). By contrast, we find that the current mean intensity values and current intensity standard deviation were poor predictors of the modification status of the k-mer (43–65% accuracy). As a control, we used the same set of features in control k-mers (i.e., those with the same sequence context, but centered in C, G, or U), finding that the features did not distinguish between m6A-modified and m6A-unmodified data sets (Supplementary Fig. 3).

To improve the performance of the algorithm, we then examined whether a combination of the features might improve the prediction accuracy, finding that the combination of the three features (base quality, mismatch, and deletion frequency) increased the accuracy of the model (88–91%) (Fig. 2e, Supplementary Table 2). We then tested whether the inclusion of features from the neighboring positions (−2, −1, +1, and +2) might further improve the model. Indeed, we find that the inclusion of neighboring features slightly improves the performance of the algorithm (accuracy = 97–99%), however, this was at the expense of increasing the number of false positives in the

control k-mer set — which do not contain the modification — (Fig. 2f, Supplementary Fig. 3), suggesting that features from neighboring positions should not be employed with this model.

We should note that our current algorithm has been trained using either 100% methylated or 100% unmethylated reads; however, in in vivo data, this will likely not be the scenario. Previous studies probing the m6A modification status in individual sites have estimated that m6A methylation in mRNAs occurs only partially, with methylation ratios ranging from 6 to 80%[28]. Therefore, we wondered whether our algorithm would be able to detect m6A modifications on mixtures of methylated and unmethylated reads. To test this, we sampled reads from both m6A-modified and unmodified data sets and mixed them in different proportions, to achieve partial methylation ratios of 0 (unmodified), 5, 10, 25, 50, 75, 90, 95, and 100% (fully modified). We find that the algorithm performance is dependent on the proportion of methylated reads (Fig. 2g); however, we find that even at 25% of methylation ratio, our algorithm predicts m6A sites with reasonable accuracy, with an area under the curve (AUC) of 0.72 (Fig. 2g).

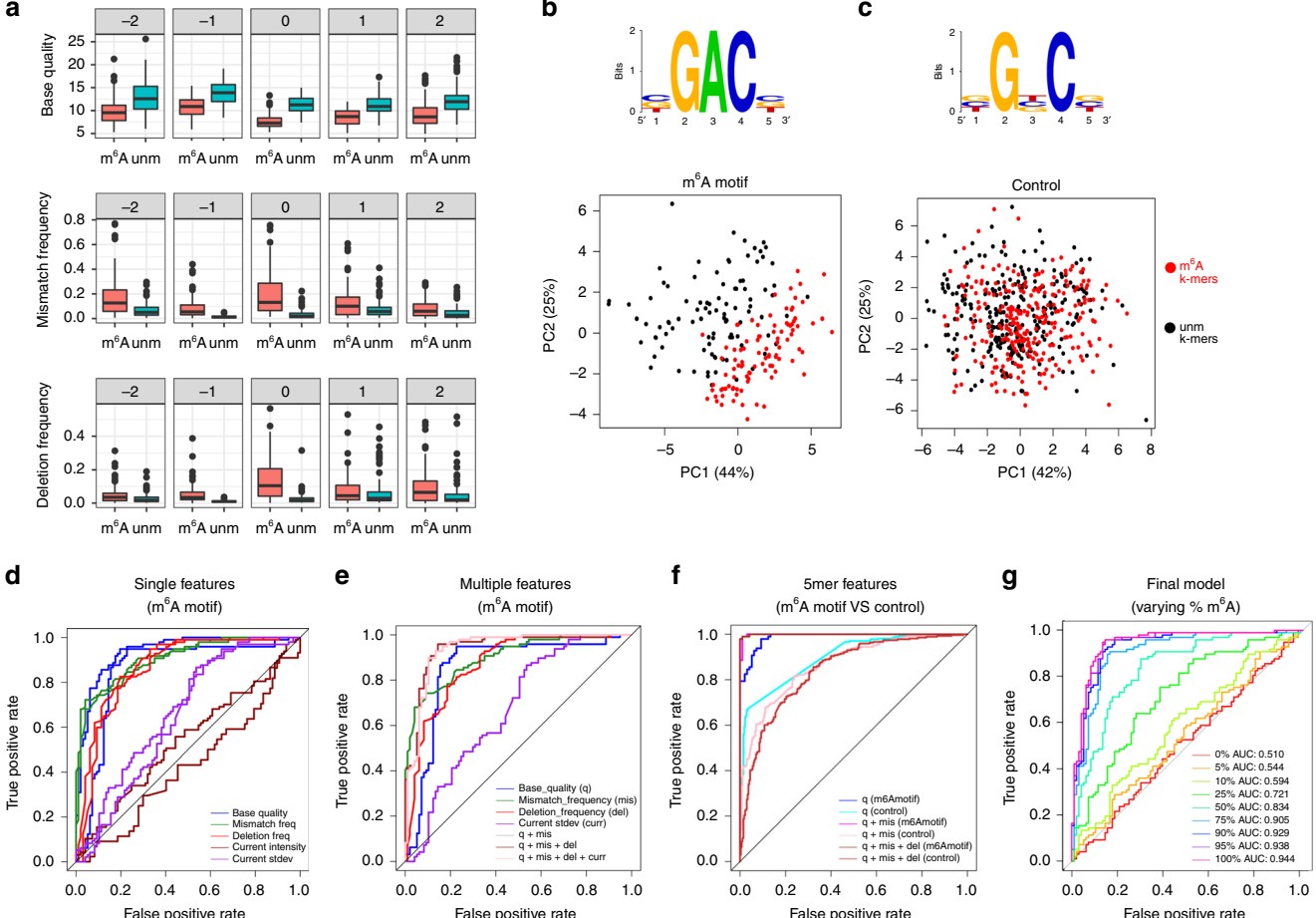

**Fig. 2** Base-calling "errors" alone can accurately identify m6A RNA modifications. **a** Base-called features (base quality, insertion frequency, and deletion frequency) of m6A motif 5-mers, and for each position of the 5-mer, are shown. The features of the m6A-modified transcripts ("m6A") are shown in red, whereas the features of the unmodified transcripts ("unm") are shown in blue. **b**, **c** Principal component analysis (PCA) scores plot of the two first principal components, using 15 features (base quality, mismatch frequency, deletion frequency, for each of the five positions of the k-mer) as input. The logos of the k-mers used in the m6A-motif RRACH set (left) and control set (right) are also shown. Each dot represents a specific k-mer in the synthetic sequence, and has been colored depending on whether the k-mer belongs to the m6A-modified transcripts (red) or the unmodified transcripts (black). The contribution of each principal component is shown in each axis. ROC curves of the SVM predictions using: (i) each individual feature separately to train and test each model, at m6A sites (**d**); (ii) combined features at m6A sites, relative to the individual features (**e**); (iii) combined features at m6A sites relative to control sites, where the base-called "errors" information of neighboring nucleotides has been included in the model (**f**); and (iv) different mixtures of methylated and unmethylated reads, using the combined features model (**g**). Error bars indicate s.d. Source data are provided in the Source Data file

**Trained SVM models can predict m6A modifications in vivo.** To assess whether our findings could be extended to in vivo data sets, we sequenced native polyA(+)-selected RNA from *S. cerevisiae* wild type (*wt*) and *ime4Δ* knockout strains (Fig. 3a and see the Methods section). *Ime4Δ* yeast strains constitute an excellent background model to identify false positives in m6A analyses[29], as the deletion of *ime4* results in complete elimination of m6A. Biological triplicates of polyA(+)-selected RNA from both *wt* and *ime4Δ* strains were sequenced in independent flow cells (see the Methods section), producing more than five million sequenced reads (Supplementary Table 3).

An initial assessment of the quality of the direct RNA sequencing runs showed that these were highly replicable both in terms of per-gene counts (spearman's rho = 0.945–0.948) and average per-read quality scores (Fig. 3b, Supplementary Fig. 4). We then used *EpiNano* to extract base-called features for all six samples. We first analyzed the features corresponding to ~1300 known m6A-modified RRACH sites, previously identified using antibody immunoprecipitation coupled to next-generation sequencing (m6A-Seq)[29]. We found that base-called features at

m6A-modified RRACH sites were distinct across yeast strains (*wt* and *ime4Δ*), for all three metrics analyzed (base quality, deletion frequency and mismatch frequency) (Fig. 3c). These results were consistent across biological replicates, and are in agreement with our observations using in vitro constructs (Fig. 1c). By contrast, this was not observed when comparing unmodified RRACH base-called features across yeast strains (Supplementary Fig. 5), suggesting that the observed differences were due to the presence of m6A. These results were further confirmed by individual inspection of "known" m6A-modified sites, where both increased mismatch and deletion frequencies were consistently observed in *wt* m6A-modified positions, but not in their corresponding *ime4Δ* sites (Fig. 3d).

To determine whether our trained SVM could be applied to in vivo data sets, we first investigated whether the global in vivo base-called features were consistent with those observed in vitro. We found that unmodified in vitro sequences (CC 0%) displayed similar mismatch frequencies to those observed in *ime4Δ* strains, which also lack m6A modifications (Fig. 3e). By contrast, m6A-modified yeast RNAs (*wt*) showed intermediate mismatch

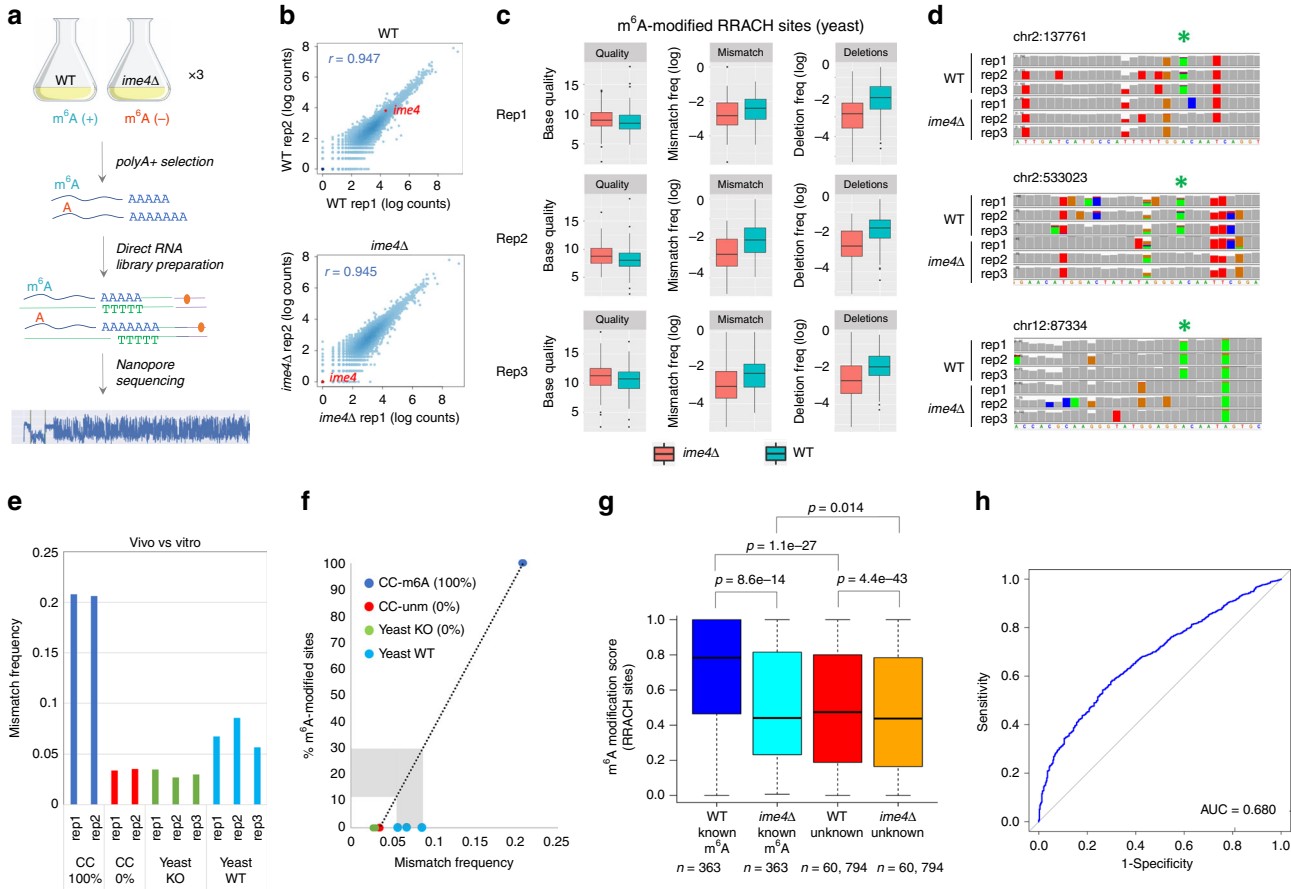

**Fig. 3** Yeast wild-type and *ime4Δ* strains show distinct base-called features at known m$^6$A-modified RRACH sites. **a** Overview of the direct RNA sequencing library preparation using in vivo polyA(+) RNA from *S. cerevisiae* cultures. **b** Replicability of per-gene counts using direct RNA sequencing across wild-type yeast strains (top) and *ime4Δ* strains (middle). The correlation between wild-type and *ime4Δ* strains is also shown (bottom). **c** Comparison of the observed mismatch frequencies in the 100%-modified in vitro transcribed sequences (blue), unmodified sequences (red), yeast *ime4Δ* knockout (green), and yeast wild type (cyan). Values for each biological replicate are shown. **d** Base-called features (base quality, insertion frequency, and deletion frequency) of RRACH 5-mers known to contain m$^6$A modifications. Only features corresponding to the modified nucleotide (position 0) are shown. Features extracted from wild-type yeast reads (m$^6$A-modified) are shown in blue, whereas those from *ime4Δ* (unmodified) for the same set of k-mers are shown in red. **f** Genomic tracks of previously reported m$^6$A-modified RRACH sites in yeast, identified using Illumina sequencing. The m$^6$A-modified nucleotide is highlighted with a green asterisk. In these positions, wild-type yeast strains show increased mismatch frequencies, as well as decreased coverage — reflecting increased deletion frequency — in all three biological replicates, whereas these features are not observed in any of the three *ime4Δ* replicates. **g** Predicted m$^6$A modification scores predicted by the trained SVM at known m$^6$A-modified (n = 363) and unknown (n = 60,794) RRACH sites, both for yeast wild-type and *ime4Δ* data sets. P-values have been computed using Kruskal–Wallis test. A site was included in the analysis if there were mapped reads present in all six yeast samples. Sites with more than one "A" in the 5-mer were excluded from the analysis. **h** ROC curve depicting the performance of *EpiNano* in yeast data sets (n = 61,363 sites). Error bars indicate s.d. Source data are provided in the Source Data file

frequencies between fully modified (CC 100%) and unmodified (CC 0%, yeast *ime4Δ*) sequences. Using linear regression, we estimated that the median stoichiometry of m$^6$A modifications in *wt* strains was 12–30% (Fig. 3f), which is in agreement with previous works, where m$^6$A was found to be present at levels ranging from 7 to 69% (with a median of 23%) in yeast samples[30]. Altogether, our results reveal that nonrandom base-called "errors" present in in vivo data sets are replicable, are in agreement with in vitro results, and are correlated with the presence of m$^6$A RNA modifications in a given site.

We then used the SVM model, previously trained with m$^6$A-modified and unmodified in vitro constructs, to predict the transcriptome-wide m$^6$A modification status of yeast RRACH sites, both in *wt* and *ime4Δ* data sets. A site was kept for downstream analyses if there was at least 1 read per site in each of the six samples. This criterion was met by 61,163 RRACH sites, from which 363 had been reported as "m$^6$A-modified", based on Illumina sequencing[29]. Per-site SVM predictions for each

biological replicate were then merged into a single "m$^6$A modification score" (see the Methods section). We should note that low read coverage leads to decreased accuracy (Supplementary Fig. 6); however, we retained low coverage sites to maximize the number of sites included in the analyses. We first compared the m$^6$A modification scores of known m$^6$A-modified RRACH sites (n = 363) in *wt* and *ime4Δ*, finding that modification scores in *wt* were significantly higher than those observed in *ime4Δ* (p = 8e−14), for the same set of sites (Fig. 3g). By contrast, modification scores of *ime4Δ* known m$^6$A-modified RRACH sites (n = 363) and unknown sites in the same strain, which do not contain m$^6$A modifications, were relatively similar (p = 0.01, Kruskal–Wallis test) (Fig. 3g). Interestingly, our method also identified significant differences in m$^6$A modification scores when comparing *wt* and *ime4Δ* unknown RRACH sites (p = 4e−43; Fig. 3g), suggesting that there might be additional m$^6$A-modified sites present in the transcriptome, apart from those identified using m$^6$A-Seq[29]. Indeed, recent efforts using enzymatic-based

m6A detection methods have reported that antibody-based methods severely underestimate the number of m6A sites[30]. Overall, we find that our model identifies m6A modifications in yeast data sets with an overall accuracy of 87.8%, recovering 32% (117 out of 363) of known m6A-modified sites (Supplementary Table 4 and Fig. 3h), and with a specificity of 89%.

**EpiNano performance compared with current intensity-based methods**. Previous efforts have attempted to identify RNA modifications from direct RNA sequencing samples by performing direct comparison of raw current intensities. This is the case of *Tombo*[22], a software originally developed for the detection of DNA modifications in nanopore sequencing data, which has recently been extended to detect RNA modifications. Identification of modifications from raw signal typically requires a two-step process: (i) re-squiggling of the raw signal to "align" all reads mapping to the same genomic location and (ii) comparison of raw current intensities across reads or samples. First, we find that the re-squiggling step used by *Tombo* discards ~50% of the reads (Supplementary Table 3). From the re-squiggled reads, we find that *Tombo* is able to identify 220 out of the 363 known m6A-modified sites in yeast *wt*, thus recovering 59.6% of known sites (Supplementary Table 4). However, this increased recovery of true positives (TP) was at the expense of increased number of false positives (*Tombo* specificity = 69.8%; *EpiNano* specificity = 89%). Thus, we find that, for the same set of 61,163 sites, *Tombo* correctly predicts known m6A sites with an accuracy of 69% and a recovery of 59%, whereas *EpiNano* predicts them with an accuracy of 87% and recovery of 32% (Supplementary Table 4).

Altogether, our in vivo analyses validate our findings using in vitro m6A-modified and unmodified sequences, and confirm the use of base-calling "errors" as a proxy to identify m6A modifications in direct RNA sequencing data sets. Furthermore, our findings validate the use of in vitro constructs, transcribed with and without RNA modifications, as a valid strategy for training direct RNA sequencing base-calling algorithms, suggesting that similar approaches could be envisioned with additional data sets containing distinct RNA modifications in the future.

## Discussion

The human epitranscriptome is still largely uncharted. Only a handful of the 170 different RNA modifications that are known to exist have been mapped. Importantly, several of these modifications are involved in central biological processes, such as sex determination[31–33] or cell fate transition[34], and their dysregulation has been linked to multiple human diseases[16,35,36], including neurological disorders[37–39] and cancers[40–42]. Yet, our understanding of this regulatory layer is restricted to a few RNA modifications, largely due to the lack of a generic methodology to map them in a transcriptome-wide fashion.

The establishment of the ONT platform as a tool to map RNA modifications has great potential to revolutionize our understanding of the epitranscriptome, as in principle, it should be capable of identifying RNA modifications in individual RNA sequences, and with single nucleotide resolution. Such ability would allow us to study the functions of the epitranscriptome in ways that, until now, have not been possible. Unfortunately, currently there is no software available that can predict RNA modifications from direct RNA sequencing reads with sufficient accuracy, limiting the applicability of direct RNA sequencing as a tool to identify RNA modifications. To tackle this limitation, here we provide a novel strategy to identify RNA modifications from base-called features, without the need of squiggling realignments or manipulation of raw current intensity data sets.

Here we report that RNA modifications can be identified in the form of systematic and reproducible base-calling "errors" in direct RNA sequencing data sets. These "errors" can be detected in the form of altered per-base qualities, mismatch frequencies and deletion frequencies at the modified site. We find that the method accurately detects modifications both in vitro (90% accuracy) and in vivo (87% accuracy), with an overall recovery of 32% of known sites. Despite the promising results, it is important to note, however, that the current method presents several limitations as well as ample room for improvement. Firstly, our current algorithm does not predict RNA modifications in individual RNA molecules, but rather employs information from all the reads mapping to a specific site to determine whether a given position is modified or unmodified. Secondly, our algorithm does not distinguish between different types of RNA modifications (e.g., m1A vs m6A). Future work will be needed to decipher whether different types of RNA modifications can be associated to distinct "error signatures", which could be potentially used to identify the underlying RNA modification type. Thirdly, although m6A-modified RRACH k-mers globally display altered base qualities, mismatch frequencies and deletion frequencies, we should note that the contribution of each feature varies across different k-mers. For example, we find that the presence of m6A in GGACA and GGACT k-mers mainly affects the mismatch and deletion frequency, whereas in the case of GGACC, base quality and deletion frequency are the most altered features by the presence of m6A modifications (Supplementary Fig. 7). Future models that include k-mer specific training and testing could potentially improve the accuracy of prediction of modified sites, as well as reduce the number of false positives. In this regard, we expect that by making our m6A-modified and unmodified data sets publicly available — both base-called fastq and raw fast5 — these can be employed by the community to train different machine learning algorithms (e.g., signal-based machine learning, base-caller training, etc.), and thus lead to improved m6A RNA modification base-callers for the whole community.

Overall, our results show that base-calling "errors" can be used as an accurate and computationally simple solution to identify m6A modifications, which does not require the manipulation of raw current intensities or squiggle alignments. Moreover, we extend our findings to an in vivo system, showing that our algorithm can capture m6A-dependent changes that are present in wild-type SK1 yeast strains, while these are not observed in their *ime4Δ* counterparts. Future work will be needed to achieve single read RNA modification detection, as well as to expand our findings to other RNA modifications.

## Methods

**Synthetic sequence design**. Sequences were designed such that they would include all possible 5-mers, while minimizing the secondary RNA structure. For this aim, we employed the software *curlcake* (http://cb.csail.mit.edu/cb/curlcake/), which internally uses RNAshapes version 2.1.6 (http://bibiserv.techfak.uni-bielefeld.de/rnashapes) to predict RNA secondary structure. The final output sequence given by the software was ~10 kb long. For synthesis purposes, a total of four sequences were designed by splitting the 10 kb sequence into smaller sequences of slightly different size (2329, 2543, 2678, and 2795 bp, which we named "Curlcake 1", "Curlcake 2", "Curlcake 3", and "Curlcake 4", respectively). Each sequence was designed with an internal strong T7 polymerase promoter, an additional BamHI site at the end of the sequence, and with all EcoRV and BamHI sites removed from the sequence (Supplementary Note 1). All four sequences were synthesized and cloned in pUC57 vector using blunt EcoRV by General Biosystems. Plasmids were double digested O/N with EcoRV-BamHI-HF, and DNA was extracted with Phenol-Chloroform followed by EtOH precipitation. Plasmid digestion was confirmed by agarose gel. Digestion product quality was assessed with Nanodrop before proceeding to in vitro transcription.

**In vitro transcription, capping and polyadenylation**. In vitro transcribed (IVT) sequences were produced using the Ampliscribe™ T7-Flash™ Transcription Kit (Lucigen-ASF3507), using 1 μg of purified digestion product as starting material,

following the manufacturer's recommendations. ATP was replaced by $N^6$-methyladenosine-5′-triphosphate($m^6$ATP) (Trilink-N-1013;) for the IVT reaction of $m^6$A-modified RNA. IVT reaction was incubated for 4 h at 42 °C. IVT RNA was then incubated with DNAse I (Lucigen), followed by purification using the RNeasy Mini Kit (Qiagen-74104). Integrity and quality of the RNA was determined using Agilent 4200 Tapestation, to ensure that a single product band of the correct size had been produced for each IVT product (Supplementary Fig. 8). Each IVT product was 5′ capped using vaccinia capping enzyme (NEB-M2080S) following the manufacturer's recommendations. The capping reaction was incubated for 30 min at 37 °C. Capped IVT products were purified using RNAClean XP Beads (Beckman Coulter-A66514). Poly(A)-tailing was performed using E. coli Poly(A) Polymerase (NEB-M0276S), following the manufacturer's recommendations. Poly(A)-tailed RNAs were purified using RNAClean XP beads, and the addition of poly(A)-tail was confirmed using Agilent 4200 Tapestation (Supplementary Fig. 8). Concentration was determined using Qubit Fluorometric Quantitation. Purity of the IVT product was measured with NanoDrop 2000 Spectrophotometer (Supplementary Table 5).

**Yeast culturing.** Sk1 strains used in this study were SAy841 comprising a deletion of NDT80 (hereafter, referred to as "wild-type"), and SAy966, in which both NDT80 and IME4 were deleted (hereafter, referred to as "$ime4\Delta$"). These strains are characterized in Agarwala et al.[43]. To induce synchronous meiotic entry, cells were grown for 24 h in 1% yeast extract, 2% peptone, 4% dextrose at 30 °C, diluted in BYTA (1% yeast extract, 2% tryptone, 1% potassium acetate, 50 mM potassium phthalate) to OD600 = 0.2 and grown for another 16 h at 30 °C, 200 rpm. Cells were then washed twice with water and resuspended in SPO (0.3% potassium acetate) at OD600 = 2.0 and incubated at 30 °C at 190 rpm. Cells were isolated from SPO following 5 h and collected by 2 min centrifugation at 3000 × g. Pellets were snap frozen and stored at −80 °C for RNA extraction. Three independent biological replicates for each strain were collected.

**Yeast mRNA preparation.** Yeast total RNA samples were prepared using a modified protocol of nucleospin® 50RNA kit (Macherey-Nagel, cat 740955.50). Specifically, cells lysis was done in a 1.5 ml tube by adding 450 µl of lysis buffer containing 1 M sorbitol (SIGMA-ALDRICH), 100 mM EDTA and 0.45 µl lyticase (10 U/µl). The sample was incubated in 30 °C for 30 min to break the cell wall, centrifuged for 10 min at 800 × g, and the supernatant was removed. From this stage, extraction proceeded as in the protocol of the nucleospin® 50RNA kit, only substituting β-mercaptoethanol with DTT. Enrichment of polyadenylated RNA from total RNA was performed using the Oligo (dT) dynabeads mRNA-DIRECT kit (Thermo Scientific, 61012) for small mRNA amounts.

**Direct RNA library preparation and sequencing.** RNA library for direct RNA Sequencing (SQK-RNA001) was prepared following the ONT Direct RNA Sequencing protocol version DRS_9026_v1_revP_15Dec2016. Briefly, 800 ng of Poly(A)-tailed and capped IVT RNA — in the case of curlcakes — or 500 ng of yeast polyA + RNA were ligated to ONT RT Adapter using concentrated T4 DNA Ligase (NEB-M0202T), and was reverse transcribed using SuperScript III Reverse Transcriptase (Thermo Fisher Scientific-18080044). The products were purified using 1.8X Agencourt RNAClean XP beads (Fisher Scientific-NC0068576), washing with 70% freshly prepared ethanol. RNA Adapter (RMX) was ligated onto the RNA:DNA hybrid, and the mix was purified using 1X Agencourt RNAClean XP beads, washing with Wash buffer twice. The sample was then eluted in elution buffer and mixed with RNA running buffer (RRB) prior to loading onto a primed R9.4.1 flow cell, and ran on a GridION (MinION for the second replicate) sequencer with MinKNOW acquisition software version v1.14.1 (v1.15.1 for the second replicate in the curlcake experiment). The sequencing was performed in independent days and machines, with two biological replicates for each "curlcake" experiment condition (nonmodified and $m^6$A-modified RNA, total of four flow cells). Each biological replicate and condition was sequenced independently in a different flow cell. For the in vivo analysis in S. cerevisiae, three biological replicates for each yeast strain (wild type and $ime4\Delta$) were sequenced, and each biological condition and replicate was sequenced in an independent flow cell (total of six flow cells).

**Base-calling, filtering, and mapping.** Reads were locally base-called using Albacore 2.1.7 (ONT). Base-called reads were filtered using NanoFilt, a component from Nanopack with settings "-q 0 -headcrop 5 -tailcrop 3", and mapped to the 4 synthetic sequences using minimap2 with the settings -ax map-ont. Mapped reads were then converted into mpileup format using Samtools version 1.4. Read basecalling and mapping metrics can be found in Supplementary Table 1 and Supplementary Table 3. For comparison, reads were also base-called with Albacore 2.3.4 and Guppy 2.3.1, finding that all base-callers showed increased mismatch frequencies in $m^6$A-modified data sets (with the largest increased in A positions) and decreased qualities (Supplementary Fig. 6).

**Feature extraction.** To extract per-site features (mean per-base quality, mismatch frequency, insertion frequency, and deletion frequency), BAM alignment files were converted to tab delimited format using sam2tsv from jvarkit. For each individual

reference site, the mean quality of the aligned bases, the mismatch, insertion, and deletion frequency was computed using in-house scripts (available on github). To extract current intensity information from individual reads, the h5py (version 2.7.0) module in python was used to parse each individual fast5 file. Reference sequences were slided with a window size of 5 bp, and mean and standard deviation of current intensities was computed for each sliding window. All in-house python scripts used to extract the features described above are publicly available as part of EpiNano (github.com/enovoa/EpiNano).

**Machine learning.** The set of extracted features of both $m^6$A-modified and unmodified "curlcakes" was used as input to train a SVM. Initial training (75% of the sites) and testing (25%) of the SVM was performed with $m^6$A-modified and unmodified curlcake reads from one replicate (rep1). Multiple kernels ("linear", "poly", and "rbf") were compared, and the best performing kernel was retained. The model was validated on new sequencing runs of IVT $m^6$A-modified and unmodified sequences (rep2), which had not been used for initial training or testing of the SVM. The reported accuracy values refer to the predictions on replicate 2. The code to extract the set of features for machine learning from fastq and fast5 reads, the code for building the SVM models, as well as the trained SVM models, are publicly available in github (github.com/enovoa/EpiNano). We should note that a limitation in utilizing IVT to generate all possible 5-mers is that 5-mers that contain more than one "A" will contain more than one modification in the k-mer, e.g., AGACC will in fact be $m^6$AG$m^6$ACC, which are unlikely to occur in a biological context. Therefore, 5-mers that contained more than one A have been excluded from the analyses, as well as from the training set. Accuracy of the model has been computed as the sum of correct $m^6$A modification predictions — correctly predicted $m^6$A-modified k-mers (TP) and correctly predicted unmodified k-mers (true negatives, TN) — divided by the total number of k-mers tested.

**Prediction of $m^6$A modified sites in yeast using EpiNano.** EpiNano was used to extract per-site features (mean per-base quality, mismatch frequency, insertion frequency, and deletion frequency) from the mapped BAM files of the six samples sequenced (WTrep1, WTrep2, WTrep3, ime4Δrep1, ime4Δrep2, and ime4Δrep3). $m^6$A-modified RRACH sites with minimum coverage of 5 reads/site were kept and scored using the previously trained SVM model. 5-mers containing more than one "A" in the motif were discarded from downstream analyses, as these k-mers had not been included in the training sets (see above). A total of 61,163 sites were analyzed for each sample and replicate, from which 363 corresponded to "known" $m^6$A-modified sites, which had been identified using Illumina sequencing[44].

$M^6$A modification scores for each site were computed by merging the SVM predicted probabilities across replicates. Specifically, if the probability being modified was >0.5 in all three biological replicates (s1, s2, and s3), the modification score (M) was set to 1. Otherwise, the modification score was determined by computing the mean of the probabilities (pseudocode 1). Modification scores were obtained for each site, both for wild-type and $ime4\Delta$ strains.

*Pseudocode 1:*
if (s1 ≥ 0.5 and s2 ≥ 0.5 and s3 ≥ 0.5):
M = 1
else:
M = (s1 + s2 + s3)/3

To classify a site as "$m^6$A-modified" or "unmodified", we compared the modification scores of each site, obtained for each of the two strains. Specifically, modification ratio was calculated by dividing the modification score of the wild-type strains ($M_{wt}$) and the modification score of the $ime4\Delta$ strains ($M_{ko}$). A site was considered to be modified if the modification ratio was >1.5 and the modification score in wild-type strains ($M_{wt}$) was greater than 0.5 (pseudocode 2).

*Pseudocode 2:*
if ($M_{wt}/M_{ko}$) > 1.5 and $M_{wt}$ > 0.5:
status = modified
else:
status = unmodified

Accuracy of the predictions was computed as the sum of correct $m^6$A modification predictions — correctly predicted $m^6$A-modified k-mers (TP) and correctly predicted unmodified k-mers (TN) — divided by the total number of k-mers tested (n = 61,163). Positive predictive value was computed by dividing the number correctly predicted $m^6$A-modified k-mers (TP) by the total number of $m^6$A-modified k-mers included in the analysis (n = 363). The performance of EpiNano on yeast wt and $ime4\Delta$ samples can be found in Supplementary Table 4.

**Prediction of $m^6$A modified sites using Tombo.** We first ran Tombo version 1.5[22] to align the raw signal and the base-called reads sequences (tombo resquiggle), both for wild type and $ime4\Delta$ samples. We then used the Tombo "canonical sample comparison" method (tombo model_sample_compare) to identify significant shifts in raw signals in paired data sets (wt and $ime4\Delta$) using the parameter -num-most-significant-stored 14,000,000 and -minimum-test-reads 1. $M^6$A modification scores for each site were computed by merging the Tombo predicted probabilities across replicates. Specifically, if the probability being modified was >0.5 in all three biological replicates (s1, s2, and s3), the modification score (M) was set to 1, as previously done for EpiNano. Otherwise, the modification score was determined by

computing the mean of the probabilities (pseudocode 1). A site was considered as modified if the modification score was >0.5. The performance of *Tombo* on yeast *wt* and *ime4Δ* samples can be found in Supplementary Table 4.

**Reporting summary**. Further information on research design is available in the Nature Research Reporting Summary linked to this article.

## Data availability

All code used in this work is publicly available at github.com/enovoa/EpiNano. All FASTQ files data generated in this work have been made publicly available at the GEO database under the accession code GSE124309 ("curlcakes") and GSE126213 (yeast wild type and ime4Δ). Fast5 raw data have been made publicly available in SRA under the accession SRP174366. All other data are available from the authors upon reasonable request.

## Code availability

The code to extract RNA modification information from direct RNA sequencing data sets, as well as all in-house python scripts used to extract the base-called features are publicly available as part of *EpiNano* (github.com/enovoa/EpiNano).

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

## Acknowledgements

O.B. is supported by an international PhD fellowship (UIPA) from the University of New South Wales. M.C.L. is supported by CRG International PhD Fellowships Programme. E.M.N was supported by a DECRA fellowship from the Australian Research Council (DE170100506) and is currently supported by CRG Severo Ochoa Funding. This work was funded by the Australian Research Council (DP180103571). We acknowledge the support of the Spanish Ministry of Economy, Industry and Competitiveness (MEIC) to the EMBL partnership, Centro de Excelencia Severo Ochoa and CERCA Programme/Generalitat de Catalunya. C.E.M thanks funding from the Bert L and N Kuggie Vallee Foundation, the WorldQuant Foundation, The Pershing Square Sohn Cancer Research Alliance, NASA (NNX14AH50G, NNX17AB26G), the National Institutes of Health (R01ES021006, 1R21AI129851, and 1R01MH117406), the Bill and Melinda Gates Foundation (OPP1151054), the Leukemia and Lymphoma Society grants (LLS 9238-16, LLS-MCL-982). We would like to thank James Ferguson for all his helpful comments, as well as for giving us early access to his fast5 processing toolkit (https://github.com/Psy-Fer/fast5_fetcher).

## Author contributions

H.L. performed the bioinformatics analysis, together with E.M.N. O.B. and M.C.L. performed the experimental work including the preparation and running of the direct RNA sequencing libraries. J.M.R. performed base-caller comparison analyses. D.W. performed the yeast culturing and mRNA purification. H.L., O.B., M.C.L., and E.M.N.

prepared the figures. E.M.N. and M.A.S. conceived the project. E.M.N. supervised the project, with contribution of S.S., C.E.M., J.M., and M.A.S. H.L., O.B., and E.M.N. wrote the manuscript, with the contribution of all authors.

## Additional information

**Competing interests:** M.S. and E.M.N. have received travel and accommodation expenses to speak at Oxford Nanopore Technologies conferences. Otherwise, the authors declare no competing interests.

