## [Peer Review File · Nature Communications]

Reviewers' Comments:

Reviewer #1:

Remarks to the Author:

The very preliminary study described in the manuscript entitled "Accurate detection of m6A RNA modifications in native RNA sequences," describes an Oxford Nanopore Sequencing technology (ONT)-based approach for identifying m6A modified ribonucleotides in in vitro transcribed RNA molecules. The approach is of interest. However, the usability of the described analytical pipeline is still completely up for debate because all analyses and bench mark tests were only performed with in vitro transcribed RNA molecules where the sequence contexts and m6A levels can and were specifically controlled to show the ability of the approach to detect m6A in a completely contrived system. There were no experiments presented to show that this approach can be used on biological samples that would be of interest to researchers in the field of post-transcriptional regulation. Until it can be proved that this technology can be used in a biologically meaningful way it is far too preliminary for publication. I also note there are some grammatical errors that should be cleaned up before this is resubmitted for publication.

Reviewer #2:

Remarks to the Author:

The paper presents a simple observation that base calling in Oxford Nanopore RNA reads makes more errors in regions that contain methylated bases than in those that contain only standard bases. The claim is that this can be exploited to detect methylated bases. The authors support their claim by performing experiments, where they synthesized sequence specifically designed to contain many relevant sequence contexts, using methylated As in one experiment and unmethylated As in a control experiment.

The work is important. "Everyone" knows that the methylation signal is contained in ONT reads, but there is little work actually exploiting this. The experimental data sets produced as part of the paper are a useful resource for further study of RNA methylation in ONT reads. However, it seems to me that the tools presented are still far from generally useable on real data, and, indeed, authors do not analyse any native RNA data sets (even though "native RNA" figures prominently in the title of the paper). Said that, I don't think that that should be held against the authors. Timely release of the paper will likely inspire further research in methylation detection from ONT RNA reads and thus the paper has a great potential to be highly cited.

Major comments:

The main problem that I have with the paper is that it is very difficult to extract details from available descriptions and these details are, in my opinion, key in the evaluation of presented results. In particular, in Results it is unclear, whether methylation detection is on individual reads, or the method is using all reads overlapping a particular position in the reference. The first problem is much more difficult.

I found the only hint in the "Feature extraction" section in Methods, where it seems to me that the features were extracted from multiple reads overlapping the same reference position and thus the classification is on the positions, not on individual reads. I will therefore assume that the authors are solving the simpler problem; either way, this should be made very clear in the presentation of results. In the rest, I assume that the authors work on the simpler problem.

Looking at many reads (how many? paper does not say!) overlapping the same position and expecting consistent methylation is, I believe, an unrealistic assumption (though it is true, that in their datasets, they should have either 100% methylated or 100% unmethylated reads). On real data sets, the results on real data are likely to be strongly dependent on the number of reads

overlapping the position, as well as percentage of methylated reads in the mixture. It seems to me that all evaluation has been done on 100% methylated or 100% unmethylated high coverage data sets and in such context, the presented extraordinary ROC curves would not be surprising.

Please, clarify the setup and make the conditions of experiment (including depth of coverage) clear. It would be great, if some experiments were also done on a mixture of methylated and unmethylated reads and a question of what percentage of methylated reads in the mixture is detectable by this method could be addressed.

You should also include a discussion on how the artificial conditions in your data sets differ from typical conditions encountered in real data analysis, outlining limitations of your approach. Some analysis on a data set from real RNA transcripts would be also great; I would not expect "gold standard" evaluation, but at least some demonstration that the method can bring some insight in case of real data.

It would be great, if you could also release the raw fast5 files in addition to the fastq files.

Finally, the standard ONT base caller changes quite quickly and the signal that you are using may disappear from the future versions of the base caller, depending on how ONT modifies their internal training set. Could you please comment on robustness of your method with respect to the base caller (or a version of the base caller) used?

Minor comments:

l. 52: modifications|transcription-wide

l. 65: the sentence starting with "These technologies" is awkward and difficult to understand.

l. 78: statement "with an overall accuracy of ~90%" seems to indicate that individual methylated As can be detected in individual reads with this accuracy. I believe this is not true (see above).

l. 243-255: any possibility of any cross-contamination of the samples, or there was a new flow cell used in each run?

l. 278: something is missing in this section?

general: please mention Tombo and compare at least briefly your methods to theirs.

Thank you for your time in considering our work and for your contributions to improve this work. Please find below our point-by-point replies to each individual concern/comment raised by the reviewers.

To address the concerns raised by Reviewer #1, we have now included *in vivo* data, which validates our finding that base-called features can be used to identify m6A modifications in direct RNA sequencing datasets, both *in vitro* and *in vivo*. More specifically, we have performed direct RNA sequencing on yeast polyA+ RNA, in biological triplicates, comparing two different strains, one that contains m6A ('wild-type') and one that does not contain m6A (*ime4Δ*). We feel that by including this additional data, the value of our methodology is further validated, and the work is greatly improved both in terms of quality and impact.

To address the concerns raised Reviewer #2, in addition to making available the code and fastq, we have also made available the raw fast5. We have also added a significant number of additional analyses and algorithm benchmarking, including testing the performance of the algorithm with different proportions of methylated/unmethylated reads, as well as with varying read coverage. We have also added an in-depth discussion on the limitations and strengths of our current method, as well as significant edits to the manuscript to clarify any points which may not have been sufficiently clear in our first submission.

Overall, we would like thank the reviewers for helping us improve the overall quality of the manuscript through their questions and concerns.

To facilitate the review process of this new submission, all edits done to the new manuscript text have been highlighted in blue.

We hope that you will find this new improved version suitable for publication in your journal, and thank you again for your time in considering our work.

Reviewers' comments:

Reviewer #1 (Remarks to the Author):

The very preliminary study described in the manuscript entitled "Accurate detection of m6A RNA modifications in native RNA sequences," describes an Oxford Nanopore Sequencing technology (ONT)-based approach for identifying m6A modified ribonucleotides in *in vitro* transcribed RNA molecules. The approach is of interest. However, the usability of the described analytical pipeline is still completely up for debate because all analyses and bench mark tests were only performed with *in vitro* transcribed RNA molecules where the sequence contexts and m6A levels can and were specifically controlled to show the ability of the approach to detect m6A in a completely contrived system. There were no experiments presented to show that this approach can be used on biological samples that would be of interest to researchers in the field of post-transcriptional regulation. Until it can be proved that this technology can be used in a biologically meaningful way it is far too preliminary for publication. I also not there are some grammatical errors that should be cleaned up before this is resubmitted for publication.

We thank the reviewer for his/her time in reviewing our work, and are glad to hear that he/she considers the work of interest. The reviewer is concerned by the usability of the methodology in *in vivo* datasets, and we have tried to address this in the current reviewed version. More specifically, we have now included *in vivo* data from yeast wild-type and *ime4* knockout (which lacks m6A) strains, finding that known m6A sites display distinct base-calling 'error' patterns in the wild-type, compared to the *ime4* knockout, in agreement with our observations *in vitro*. Overall, we now show that our methodology is applicable to biological samples, with an accuracy of 87.8%, a specificity of 89% and a recovery of 32%.

We agree with the reviewer when he/she mentions that the inclusion of *in vivo* data would greatly improve the quality and impact of this work; however, we would like to note that in our opinion, the value of our work is not only the applicability of our methodology to genome-wide *in vivo* datasets. We believe that an important contribution of our work is the realisation that RNA modifications (in this case, m6A) can be detected in the form of base-calling errors, which has not been reported to date (all other previous efforts have so far been exclusively focused on the comparison of raw signal current intensities, such as the ONT software Tombo, which we find performs poorly in our datasets). Therefore, we are proposing a completely novel strategy, different from previous efforts, which can be used to detect RNA modifications in direct RNA sequencing datasets. Importantly, having more metrics than

just current intensity to detect RNA modifications will likely pave the path for future efforts aiming to distinguish among different types of RNA modifications (e.g. m1A vs m6A). We have now included all these comments, as well as a broader description of the current limitations and possible improvements of the methodology, in a new paragraph in the Results and Discussion section.

Reviewer #2 (Remarks to the Author):

The paper presents a simple observation that base calling in Oxford Nanopore RNA reads makes more errors in regions that contain methylated bases than in those that contain only standard bases. The claim is that this can be exploited to detect methylated bases. The authors support their claim by performing experiments, where they synthesized sequence specifically designed to contain many relevant sequence contexts, using methylated As in one experiment and unmethylated As in a control experiment.

The work is important. "Everyone" knows that the methylation signal is contained in ONT reads, but there is little work actually exploiting this. The experimental data sets produced as part of the paper are a useful resource for further study of RNA methylation in ONT reads. However, it seems to me that the tools presented are still far from generally useable on real data, and, indeed, authors do not analyse any native RNA data sets (even though "native RNA" figures prominently in the title of the paper). Said that, I don't think that that should be held against the authors. Timely release of the paper will likely inspire further research in methylation detection from ONT RNA reads and thus the paper has a great potential to be highly cited.

We thank the reviewer for his/her time in reviewing our work, and are glad to hear that he/she considers the work important and of interest for the community. We agree that our datasets can be a useful resource to study RNA methylation in ONT reads, and have accordingly made available the raw fast5 reads, such that others can reuse our raw data to train and test their own algorithms.

We agree with the reviewer that our research will likely enhance the development of novel strategies for methylation detection from ONT reads. Indeed, we envision that future approaches could in fact use the base-called 'error' signatures described in this work in combination with other strategies, as we have now mentioned in the Discussion.

With regards to the title, we originally employed the term "native RNAs" because we are sequencing the RNA molecules directly, rather than sequencing cDNA, in contrast to current m6A-Seq methodologies used to detect m6A sites. That being said, we agree with the reviewer that this term could be misleading. However, considering that now we are also including *in vivo* data in the current version of the manuscript, we have left the title as in the original submission.

Major comments:

The main problem that I have with the paper is that it is very difficult to extract details from available descriptions and these details are, in my opinion, key in the evaluation of presented results. In particular, in Results it is unclear, whether methylation detection is on individual reads, or the method is using all reads overlapping a particular position in the reference. The first problem is much more difficult.

We have now clarified throughout the text that we are identifying methylated sites by using all reads overlapping to a particular position in the reference. As the reviewer mentions this problem is much more difficult, and unfortunately our algorithm is still not yet sufficiently accurate to properly detect modifications at the level of individual reads. We have clarified this now in the text and have added a paragraph in the discussion where we discuss all the limitations and possible improvements of the current proposed methodology.

I found the only hint in the "Feature extraction" section in Methods, where it seems to me that the features were extracted from multiple reads overlapping the same reference position and thus the classification is on the positions, not on individual reads. I will therefore assume that the authors are solving the simpler problem; either way, this should be made very clear in the presentation of results. In the rest, I assume that the authors work on the simpler problem.

We thank the reviewer for noting this, and we have now clarified this in the main text, in addition to being described in the Methods section. Furthermore, this has also been clarified in the github release.

Looking at many reads (how many? paper does not say!) overlapping the same position and expecting consistent methylation is, I believe, an unrealistic assumption (though it is true, that in their datasets, they should have either 100% methylated or 100% unmethylated reads). On real data sets, the results on real data are likely to be strongly dependent on the number of reads overlapping the position, as well as percentage of methylated reads in the mixture. It seems to me that all evaluation has been done on 100% methylated or 100% unmethylated high coverage data sets and in such context, the presented extraordinary ROC curves would not be surprising.

With regards to the reviewer's question of how many reads are used, this information can be found in Table S1. We have now cited this Table in the main text (and not only in the Methods section as in the original submission) to make sure that the reader can easily find this information. In addition, the processed data showing the individual coverage at each individual site can be found in the GEO release in the form of processed supplementary data.

With regards to the reviewer's comment about methylation percentage, our *in vitro* datasets are either 100% methylated or 100% unmethylated, as the reviewer mentions. Therefore, the reviewer is correct when he/she mentions that on real datasets, this will likely not be the case. Previous studies characterizing the m6A stoichiometry in individual sites have estimated that m6A methylation occurs with partial methylation ratio ranging from 6% to 80% (Liu et al., RNA 2013). Nevertheless, here we now show that we are able to identify differences in the base-called features of known m6A sites when comparing yeast wild-type versus ime4Δ strains, despite the fact that yeast mRNAs are not 100% methylated.

Please, clarify the setup and make the conditions of experiment (including depth of coverage) clear. It would be great, if some experiments were also done on a mixture of methylated and unmethylated reads and a question of what percentage of methylated reads in the mixture is detectable by this method could be addressed.

We have now clarified the conditions of the experiment throughout the manuscript. The per k-mer coverage for each individual site is included in the processed data that is uploaded in GEO release GSE124309 (supplementary file GSE124309_RAW.tar). We have also now included a new Figure S6, where distribution of per-site coverage is shown in the yeast datasets.

With regards to the reviewer's suggestion of using mixtures of methylated/unmethylated reads, we have now included a simulation of how our trained SVM performs on sites with different proportions of m6A modified reads, and have now included this information as panel G in Figure 2.

In addition, we have also included *in vivo* data which is in fact a natural "mixture", where we find that known m6A sites display distinct base-called features between the two strains (wild-type and ime4Δ).

You should also include a discussion on how the artificial conditions in your data sets differ from typical conditions encountered in real data analysis, outlying limitations of your approach. Some analysis on a data set from real RNA transcripts would be also great; I would not expect "gold standard" evaluation, but at least some demonstration that the method can bring some insight in case of real data.

Following the reviewer's suggestion, we have now included an in-depth discussion on the limitations of the current method, as well as potential improvements (see highlighted paragraphs in "Discussion").

In addition, following the reviewer's suggestion, we have now included *in vivo* data, where we find that known m6A sites display distinct features in wild-type versus ime4Δ strains, in biological triplicates, showing that in real scenarios of "mixture" of modified and unmodified reads/transcripts, differences in base-called features can still be observed and can be used to detect m6A modifications with reasonable accuracy (Figure 3).

It would be great, if you could also release the raw fast5 files in addition to the fastq files.

We have now made this available, and have uploaded them to SRA, under the accession SRP174366. We thank the reviewer for pointing this out.

Finally, the standard ONT base caller changes quite quickly and the signal that you are using may disappear from the future versions of the base caller, depending on how ONT modifies their internal training set. Could you please comment on robustness of your method with respect to the base caller (or a version of the base caller) used?

The reviewer is right with regards to his/her comment on the potential dependency on the base-caller. To ensure that our code can be used in the future without problems, we have included the specific albacore release in github, under the section 'third-party' softwares. We have also included supplementary data with regards how using different base-callers (Guppy vs Albacore) and different versions of the same base-caller (Albacore version 2.1.3 versus 2.3.4), systematically affect the base-called features (mismatch frequency, deletion frequency and per-base quality), regardless of the base-caller used (Figure S9).

Minor comments:

I. 52: modifications|transcription-wide

This has now been corrected. We thank the reviewer for highlighting this typo.

I. 65: the sentence starting with "These technologies" is awkward and difficult to understand.

We agree with the reviewer that this sentence was difficult to understand, and we have rephrased this sentence accordingly.

I. 78: statement "with an overall accuracy of ~90%" seems to indicate that individual methylated As can be detected in individual reads with this accuracy. I believe this is not true (see above).

We have rephrased this sentence, as we agree that the use of "native" in this context can be misleading. Here we referred to native RNAs in the sense that they are not reverse transcribed products, i.e. in the "original" (or "native") molecule. We agree with the reviewer that this term can lead to confusion, and therefore have removed the word "native" from the sentence.

I. 243-255: any possibility of any cross-contamination of the samples, or there was a new flow cell used in each run?

Each sample and biological replicate was run in a different flowcell, thus eliminating the possibility of cross-contamination of samples. A total of 4 flowcells were run for the curlcake datasets, and a total of 6 flowcells were run for the yeast datasets. We have now clarified this in the methods section "Direct RNA library preparation and sequencing".

I. 278: something is missing in this section?

We are not sure what the reviewer refers to here. Nevertheless, we have rephrased this sentence to make it clearer.

general: please mention Tombo and compare at least briefly your methods to theirs.

While the Tombo *bioRxiv* paper was cited in our original submission, we had not included a comparison of this algorithm to our own. We have included the results of Tombo in the manuscript, and have compared and discussed the performance of both methods (Table S4).

Reviewers' Comments:

Reviewer #1:

Remarks to the Author:

The authors have adequately addressed all of my concerns. I do not have a minor editorial issue that needs to be corrected in the published article which is that m⁶A should be denoted as m⁶A not the non-superscripted form of the designation.

Reviewer #2:

Remarks to the Author:

All issues from my previous report have now been satisfactorily addressed by the authors.

REVIEWERS' COMMENTS:

Reviewer #1 (Remarks to the Author):

The authors have adequately addressed all of my concerns. I do note a minor editorial issue that needs to be corrected in the published article which is that m6A should be denoted as m⁶A not the non-superscripted form of the designation.

We thank the reviewer for his/her time of reviewing this work.

Following the reviewer's suggestion, we have changed this designation in the revised version of the manuscript.

Reviewer #2 (Remarks to the Author):

All issues from my previous report has now been satisfactorily addressed by the authors.

We thank the reviewer again for his/her time of reviewing this work.